# Getting to the Meat of It: The Effects of a Captive Diet upon the Skull Morphology of the Lion and Tiger

**DOI:** 10.3390/ani13233616

**Published:** 2023-11-22

**Authors:** David M. Cooper, Nobuyuki Yamaguchi, David W. Macdonald, Bruce D. Patterson, Galina P. Salkina, Viktor G. Yudin, Andrew J. Dugmore, Andrew C. Kitchener

**Affiliations:** 1Department of Natural Sciences, National Museums Scotland, Edinburgh EH1 1JF, UK; a.kitchener@nms.ac.uk; 2Institute of Geography, School of Geosciences, University of Edinburgh, Edinburgh EH8 9YL, UK; andrew.dugmore@ed.ac.uk; 3Institute of Tropical Biodiversity and Sustainable Development, University Malaysia Terengganu, Kuala Terengganu 21030, Malaysia; 4Wildlife Conservation Research Unit, Department of Zoology, University of Oxford, The Recanti-Kaplan Centre, Tubney House, Abingdon Road, Abingdon OX13 5QL, UK; david.macdonald@biology.ox.ac.uk; 5Negaunee Integrative Research Center, Field Museum of Natural History, 1400 S. DuSable Lake Shore Drive, Chicago, IL 60605, USA; bpatterson@fieldmuseum.org; 6Joint Directorate of the Lazovsky State Nature Reserve and the National Park «Zov Tigra», Tiger Protect Society, Primorskij Kraj, Vladivostok 692609, Russia; tpsrus@mail.ru; 7Federal Scientific Centre for the Biodiversity of Terrestrial Biota of East Asia, Far Eastern Branch, Russian Academy of Sciences, Primorskij Kraj, Vladivostok 690022, Russia; vudin75@yandex.ru; 8Human Ecodynamics Research Center and Doctoral Program in Anthropology, City University of New York (CUNY), New York, NY 10017, USA

**Keywords:** *Panthera*, linear morphometrics, captivity, wild, phenotypic plasticity, ontogeny, development, conservation, welfare, taxonomy

## Abstract

**Simple Summary:**

This study finds that the skulls and mandibles of lions and tigers in predominantly European zoos differ in shape, but not size, from lions and tigers in the wild. The nature of the shape change found indicates that the mechanical influences of diet have influenced development. The majority of captive big cats used in this study have been fed partial or whole carcasses, which better replicate the mechanical properties of wild diets than softer prepared diets. We therefore speculate that additional mechanical stresses upon the skull and mandible such as the killing bite, manipulation such as dragging, and consumption of large prey in the wild have driven differentiation between the skulls of captive and wild big cats. It is important to understand these differences to better care for captive animals, enhance their survival in reintroductions, and to understand the importance of environmental factors in influencing an animal’s size or shape.

**Abstract:**

Zoo animals are crucial for conserving and potentially re-introducing species to the wild, yet it is known that the morphology of captive animals differs from that of wild animals. It is important to know how and why zoo and wild animal morphology differs to better care for captive animals and enhance their survival in reintroductions, and to understand how plasticity may influence morphology, which is supposedly indicative of evolutionary relationships. Using museum collections, we took 56 morphological measurements of skulls and mandibles from 617 captive and wild lions and tigers, reflecting each species’ recent historical range. Linear morphometrics were used to identify differences in size and shape. Skull size does not differ between captive and wild lions and tigers, but skull and mandible shape does. Differences occur in regions associated with biting, indicating that diet has influenced forces acting upon the skull and mandible. The diets of captive big cats used in this study predominantly consisted of whole or partial carcasses, which closely resemble the mechanical properties of wild diets. Thus, we speculate that the additional impacts of killing, manipulating and consuming large prey in the wild have driven differentiation between captive and wild big cats.

## 1. Introduction

Diet plays a key role in influencing skull morphology [1], and dietary differences have been identified as a driving force of morphological variation between captive and wild vertebrates [2], with significant implications for how animal populations are viewed and managed. Big cats, in particular, have often featured in observations and studies of differences between wild and captive individuals [3,4,5]. Even though felids are obligate carnivores, the meat available in the wild and captivity can be fundamentally different, and consumed on its own or as a partial or whole carcass with connective tissues, skin and bone. Bone as a material is phenotypically plastic, and hence the size and shape of bony structures are influenced by multiple environmental factors during life [6,7]. Full skeletal growth in lions (*Panthera leo*) is reached by 3–4 years of age [1] and variation in the forces acting on the skull because of biting, holding and chewing prey, may impact the development of skull size and shape during growth as well as in adults. Diets in zoos and wildlife parks may differ considerably from those in the wild so it is not surprising that differences in skull shape and size should occur. Here, we assess these differences in detail.

Understanding the driving forces behind morphological differentiation caused by captivity has direct welfare implications for ex situ care and has consequences for conservation and subspecific taxonomy. The IUCN One Plan approach to species conservation promotes the conservation of endangered species by managing both wild and captive populations together, which highlights the importance of ex situ populations for in situ management [8,9]. For example, a Global Species Management Plan (GSMP) was established in 2016 for the management of wild and captive populations of the Sumatran tiger. Therefore, it is important to understand the driving forces and potential impacts on morphological differences between captive and wild tigers (*Panthera tigris*) and lions, if both species are to be successfully managed as part of the IUCN One Plan approach. Morphology influences function and therefore determines fitness, which is important if captive individuals are to be reintroduced into the wild [10].

The subspecific taxonomy of big cats is controversial. Using similar data, different authors recognise between two and nine subspecies of the tiger [11,12]. Such differences in opinion could impact both in situ and ex situ conservation for this species. Many putative subspecies are either extinct or have exceptionally low populations, which may affect their future viability [13]. The revised taxonomy of felids produced by the IUCN Cat Specialist Group used the concordance of morphological, genetic and biogeographical differences to identify subspecies [14], but this assumed a dominant role of evolutionary history in determining differences in skull size and shape. Morphological variation driven by environmental factors may require revisions to this taxonomy. Recent studies have highlighted the utility of captive/wild comparison studies for understanding phenotypically plastic morphological differences between populations of wild animals, and thus the criteria for subspecific taxonomy [15,16]. Boundaries of subspecies, recognised and defined by morphological differences, influence conservation management and so have real, practical importance for the future of these species. In turn, subspecies are recognised internationally as a unit of conservation, which is recognised in national and international legislation, so clarification of subspecific taxonomy is vital for effective conservation action. 

Differences between both pelage colouration and skull shape and size of captive and wild lions were described during the early 20th century [3]. Relative to wild lions and tigers, captives have been shown to have greater rostral and mastoid breadths, and broader bizygomatic arch widths and mandible widths, yet have reduced mandible heights, and shorter overall skull lengths [3,5,17]. In American zoos, captivity has been shown to have a greater influence than sex on the craniometric shape of lions and tigers [5]. Zoos differ significantly from wild environments in available space, climate, veterinary and keeper care, diet, and species interactions. Diet is the primary influence behind differing skull morphologies between captive and wild vertebrates due to differences in their nutritional and mechanical properties [2]. The nutritional value of zoo diets can impact skeletal size and cranial shape in mammals [18,19], yet it is the interaction between teeth and jaws and the mechanical properties of food during biting and chewing that have been widely hypothesised as driving the differences in skull shape between wild and captive big cats [2,3,16,20]. This is because wild lions have been shown to possess greater skull dimensions in areas of higher stress during biting and chewing compared to captive lions, which may eat softer foods requiring less force production during chewing and manipulation [17]. Soft diets have been shown to reduce mid-palatial suture growth, narrow the premaxillae and frontal bones, decrease the mass of the cranium and mandible, and decrease the length of the angular process of the mandible in laboratory studies of the rat (*Rattus norvegicus*) [21,22,23,24,25]. Captive lion skulls, whilst more massive for a given length, weigh less than those of their wild counterparts due to the softer, more spongy nature of the bone [20]. Following weaning, which occurs between four and nine months [26,27], lions and tigers are commonly fed processed meat diets in North American zoos [28]; although these diets are nutritionally complete, they are structurally unnatural, lacking bone, skin, connective tissues and organs, and require minimal forces to chew and consume [5]. Conversely, captive felids in European institutions are commonly fed whole or partial carcasses [29,30,31], which likely better replicate the mechanical properties of wild diets. Tough diets, which include bone and skin, can increase tooth microwear in big cats, and leave recognisable signatures in their dental enamel [32]. Whilst carnivores in captivity may process long bones in their food to a greater extent than those in the wild due to stereotypic behaviours [33,34], both lions and tigers in captivity have been shown to have greater build-up of calculus on the cheek teeth and a higher prevalence of periodontal disease compared to wild specimens due to the lack of abrasion caused by soft diets [35]. In extreme cases, processed diets may cause poor development of the jaws, leading to misalignment and crowding of teeth. In captive cheetahs, *Acinonyx jubatus,* and other felids, focal palatitis (also called focal palatine erosion) may occur where the lower molars impact the palate, often causing pathological change [36,37,38]. Although this condition occurs in the wild, it is more frequent in captivity [37]. Poorly aligned jaws could affect the ability of big cats to kill their prey and consume meat.

The nutritional properties of diet, including the quantity and consistency of food throughout ontogeny, are likely to influence skull morphology in felids. Calorific intake, fat and protein composition, and vitamin/mineral deficiencies have all been shown to impact skull morphology in other mammals [39,40,41,42,43]. Protein and fat digestibility can vary by the type of processing applied to processed food [44], and between dietary items [45]. The skulls of lions in captivity may be larger and attain adult size faster due to higher levels of better nutrition compared to their wild conspecifics [1]. Captive Iberian lynxes (*Lynx pardinus*) are known to have larger body weights than wild lynxes due to a combination of increased energy intake, reduced energy requirements and changes to metabolic programming during development [10]. 

Our analysis is based on substantially larger datasets than previous craniometric studies that compared captive and wild specimens; it includes measurements from both wild and captive lion and tiger specimens, representing their entire recent historical ranges. We consider the extent to which the mechanical and nutritional properties of diet influence differences in skull morphology between captive and wild big cats. By comparing the fundamentally different environmental conditions between captive and wild lifestyles, this paper investigates the relative importance of environment compared to genetics in determining skull morphology. This has important implications for ex situ welfare, subspecific taxonomy and conservation. This new evaluation is structured around assessing the differences in skull and mandible morphology of the lion and tiger. By considering sex (owing to high degrees of sexual size dimorphism) and geographical origin, we can test if there are similar patterns of differentiation between captive and wild tigers and lions with independent datasets, to see if our results have broader implications for understanding variation in skull morphology in other species of conservation importance.

## 2. Approach and Methods

### Morphometric Methods

We utilised the crania and mandibles of lions (n = 500) and tigers (n = 389) from museum collections across the world. Captivity status (captive or wild) was determined by museum record information. We measured captive specimens that originated from zoos in Europe (n = 83), Africa (n = 23), Asia (n = 18) and from unknown captive origin (n = 18). From our sample, we removed 131 subadults due to incomplete skull development. Subadults were defined by the cemento-enamel junctions of all canines occurring above the alveoli of a cleaned skull, whilst the basioccipital-basisphenoid suture and/or frontal suture are still open [46]. 

We follow the data collection methods of [46] and analytical methods of [16] for all linear morphometric measurements and cranial volume. For each specimen, 74 linear measurements and cranial volume were recorded (see Appendix A); 19 of these measurements were discarded due to low measurement repeatability (assessed by [16] based upon the mean coefficient of variation exceeding 1% for intra- or inter-observer error), leaving 56 measurements for the analysis. Owing to skull and mandible damage and missing elements, our dataset contained some missing values. Specimens with over 20% missing data, or missing entire mandibles, were removed from the dataset (n = 135). We used multiple imputations by chained equations (MICE [47]) to account for missing data in the dataset (3.4% of all measurements), which maximised the number of specimens available for analysis. Following imputation, six specimens were removed where either captivity status or sex was unknown. Our dataset incorporates 172 measured specimens of adult continental tigers (specimens which have originated from mainland Asia) as reported by [16]. The final dataset utilised here consists of 56 measurements of 617 specimens (Figure 1 and Appendix A).

All analyses were performed in R [48]. Measurements were log-transformed, centred (by subtracting the mean value), and multiplied by a matrix of values corresponding to –(1/the number of columns of the variables) [49,50], so that measurements were independent of size. The geometric mean of all variables for each individual was calculated to provide a measure of isometric size (isosize). Isosize provides a useful metric for comparison between captive and wild specimens, as skull size is commonly used as a proxy for body size [51]. We used Principal Component Analysis (PCA) on the scaled variables to create shape Principal Components (sPCs). The relationships between the scaled variables, sPCs and isosize were examined in relation to captivity status. We highlighted variables, which differed significantly between captivity and the wild using *t*-tests (Bonferroni correction for *p*-value of 0.0009). 

The data for each sex were analysed separately for both lions and tigers due to a high degree of sexual dimorphism in big cats [1,2,52]. Owing to known geographical variation in both size and shape in the lion and the tiger [11,16,46,53,54,55,56], where sample sizes allowed, we further split the available data into groups of similar geographical origin, pertaining to putative subspecies. The data were centred and rescaled by each data subset before re-performing analyses. Our dataset consists of considerably more wild specimens than captive specimens, especially for lions. The geographical origin of captive and wild specimens is not random (Figure 1), and certain geographical groups are represented in greater or lesser numbers depending on captivity status. Captive tigers within the dataset principally consist of specimens recorded as Amur tigers and Sumatran tigers, both of which are also well represented by wild specimens. Many wild Javan tigers and Bengal tigers are available, but these groups are poorly represented by captive specimens. The lion dataset chiefly consists of wild specimens from East Africa, although there are very few captive specimens from this origin. Owing to the paucity of captive data for the lion, Asian and West African lions are analysed together, which despite their geographical separation, are classified as members of the same subspecies *P. leo leo* [14]. Owing to the well-documented differences in foramen magnum height and cranial volume between captive and wild lions, which may be related to unique captive behaviours or conditions [57], PCAs are performed on data without these two variables to highlight other differences between captive and wild specimens of male and female lions and tigers, and of nominal geographical classes.

## 3. Results

### 3.1. All Data

There is size-related sexual dimorphism in both the lion and tiger, with males being larger, but there is greater size overlap between males and females in the tiger than in the lion (Figure 2a). Shape principal component 1 (sPC1) (25.3% contribution) distinguishes between the lion and the tiger with little overlap between each species of the same sex. This component also accounts for size-related shape change (allometry) for each species. sPC2 (22.7% contribution) separates males and females of both species, although there is considerable overlap. Captive individuals differ from wild individuals across sPC3 (6% contribution), although there is considerable overlap. No relationship was found between shape principal components or isosize and specimen age (for captive specimens of known age) (Appendix A). 

### 3.2. Separation by Sex and Species

For subsequent analysis of partitioned data, the foramen magnum height and cranial volume were removed from shape PCAs due to the very high contribution of these variables to the overall variance (Appendix A), and due to their known discriminatory power between captive and wild specimens of lions [57]. Male and female lions show some differentiation by captivity status across sPC1 (14.9% and 17.5% contributions, respectively), and male and female tigers across sPC2 (13.6 and 13.6% contributions, respectively) (Figure 3 and Figure 4). In sPC1 for the male and female lion, increases in measurements of the neurocranium are negatively correlated with measurements of the teeth and certain measurements of the mandible, such as mandible height of the angular process (Appendix A). The pattern is less clear for male and female tigers.

Patterns of variation were visualised by highlighting measurements, which differ significantly between captive and wild lions and tigers (Figure 5). Statistical differences determined by *t*-tests (Appendix A) are displayed in either orange (captive) or blue (wild), depending on which measurement mean is largest. In both the lion and tiger, measurements of overall skull length are not consistently affected by captivity status, which is concordant with plots and *t*-tests showing no differentiation between captive and wild by isosize. Because the variables are scaled by isometric size, large measurements of skull length exhibit low variance. Rostral depth and breadth measurements are generally larger in captive specimens. 

Cranial volume is significantly smaller in captive specimens, but skull breadth and height measurements are generally larger, especially in the lion. Foramen magnum height and occipital condyle breadth are smaller in captive specimens, especially in the lion. There is no difference in foramen magnum height between captive and wild tigers, although there is a large variation in this measurement in both captive and wild specimens. Consistent with captive skulls being broader, measurements of the palate tend to be broader, especially in lions, which also exhibit longer palate length. Both maxillary and mandibular tooth lengths are generally reduced in captive specimens. 

Whilst mandible length from the angular process, mandible depth-I and width of the mandibular condyles are generally larger in captivity, mandible height measurements are consistently smaller. 

Measurements of overall skull length, orbit, overall facial length, palate–inion length, overall zygomatic length, cranial heights and skull heights are largely unaffected by captivity. There is no difference in postorbital bar length between wild and captive skulls, but this measurement is of interest due to its relatively large variance in both captive and wild specimens (Appendix A) and its role as an attachment site for the masseter muscle used in chewing with the premolars and molars.

### 3.3. Separation by Geographical Origin

As found by [16], in analyses of the continental tiger, it is apparent that the geographical origin of specimens may influence the analysis. For example, most captive female lions are Asian lions, and although these are separated from wild Asian lion specimens, this may affect the clarity of the results. Given the large number of captive and wild tiger specimens from the Russian Far East (Amur) and Sumatra, there is less chance of bias due to geographical origin. It is of interest that captive Amur and Sumatran tigers group together across sPC2, away from their wild counterparts, despite their very different geographical origins, genetics, taxonomic status, and isometric size (Figure 4b,d)—this supports the notion that sPC2 shows variation as a result of captivity status, yet there is still large overlap when the data are considered as a whole. Therefore, we separated specimens into groups of similar geographical origin or taxonomic status.

There is variation in skull dimensions dependent on geographical origin, as represented by the recorded putative subspecies of each specimen (Figure 3 and Figure 4). Figure 6 shows the separate sPCAs and isosizes for male and female Amur tigers and Sumatran tigers, and northern lions (Asian and West African lions). The results of the Amur tiger here are reproduced from [16] for comparison of general patterns with those of tigers from Sumatra and with Asian/West African lions. There is no difference in size between captive and wild specimens at this scale of analysis (or at coarser scales of investigation). Tigers are distinguishable by captivity status across sPC1 in female and male Amur tigers (23% and 25% contribution, respectively) with very low overlap. sPC1 separates wild and captive female Sumatran tigers (34.7% contribution) and sPC2 separates wild and captive male Sumatran tigers (19.9% contribution), although there is greater overlap between captive and wild Sumatran tigers compared with Amur tigers. Captive and wild specimens of the northern lion (Asian and West African) are distinguishable (with some overlap) for males and females across sPC1 and sPC2, but it is apparent that this is at least in part due to geographical origin, between Asia and West Africa, rather than captivity status. 

Measurements differ between captive and wild specimens in a similar way for each geographical group, as was found in the analyses of male and female lions and tigers (Appendix A). One exception to this is the significant differentiation between sagittal crest length and cranial heights of male Amur tigers, which does not separate captive from wild specimens for any other grouping. Captive male Amur tigers have significantly smaller sagittal crest lengths and cranial heights (which account for sagittal crest height). This pattern of differentiation is not apparent in female Amur tigers, tigers from Sumatra, or Asian/West African lions.

## 4. Discussion

When the data are analysed together, species (sPC1) and sex (sPC2) account for similar levels of variation, and males are consistently larger than females in both species. sPC1 also strongly separates the sexes in both species, which is likely due to allometric scaling given the size disparity between males and females. sPC3 appears to differentiate between captive and wild specimens, although there is considerable overlap, and this component accounts for only 6% of the variation compared to ~20% for sPC1 and ~20% for sPC2. Whilst big cats can live longer in captivity than in the wild [58,59], we find no relationship between adult age and skull shape or size in captive lions or tigers of known age (Appendix A), and so it is unlikely that age discrepancies between captive and wild populations have influenced our findings. Whilst previous studies have concluded that captivity status is nearly twice as discriminating as sex in craniometric studies of the lion and tiger [5], the results presented here find sex is over three times more discriminatory than captivity status. This difference may be due to the North American source of captive specimens utilised in previous analyses, as opposed to the greater geographical range of specimens used here. There is a widespread use of ground meat diets for feeding captive lions and tigers in North American institutions, and this diet lacks the mechanical properties of a natural diet, which would affect jaw musculoskeletal development [5,28,60]. Alternatively, differences in the level of discrimination of captivity status and sex may be due to the removal of Principal Component 1 from the previous analysis. This step removes the allometric scaling influence, which differentiates male and female specimens based on size. The majority of captive specimens in this study originate from European zoos, which typically serve whole or partial carcasses to their big cats rather than processed foods [29,30,31]; these better replicate the mechanical properties of a wild diet. The same style of feeding would have been prominent even for the older captive specimens in this analysis, which date from the mid-19th century in captive environments that were comparatively barren and unnatural [3]. However, it is likely that some individuals would have been fed skinned meat off the bone or with little bone or skin regardless of the time period, or have been hand-reared beyond weaning age and therefore not fed carcasses during critical years of skeletal development.

When specimens of the same taxon were analysed, the differentiation between captive and wild specimens was markedly more apparent, often accounting for the first principal component with very little overlap in shape space. Because patterns of shape variation are similar between Sumatran tigers, Amur tigers, and northern lions (Appendix A), it is unlikely that founder effects in the captive populations of each of these disparate groups have influenced our conclusions. Further, modern captive breeding programmes aim to equalise founder genetic contributions and minimise inbreeding [61,62], and historic captive specimens are most likely to have been acquired from the wild directly or be from one or relatively few captive generations. We therefore postulate that it is phenotypic plasticity in response to environmental differentiation (through diet), rather than genetic differentiation, that is the primary driver of the shape changes between captive and wild lions and tigers described here. The lack of overlap in the principal component analyses of geographically and taxonomically similar specimens shows that the effects of a captive lifestyle are still apparent, despite many European zoos utilising enrichment practices and carcass feeding, which better replicate natural forces acting on the skull and reduce stereotypic behaviours [60,63]. The wider skull dimensions and shorter mandible heights of captive specimens have been found in previous studies and likely relate to differences in forces acting on the skull during development, owing to differences in how captive and wild big cats interact with diet. This includes the mechanical properties of diet (see [64] for a comprehensive definition), as suggested by previous work on felids in captivity [3,5,16,18]. However, it is likely that the intensity of prey immobilisation, killing technique, carcass dragging and general manipulation in the wild are not well replicated in captivity. For example, after killing large prey, wild Amur tigers drag carcasses an average of 165 m (n = 20) from the kill site before consumption [65]. It is probable that these actions in the wild place additional loadings upon the skull and mandible, which influence the shape and would affect, for example, the occipital condyles, which anchor the skull to the vertebral column. Whilst European zoos may provide whole carcasses of small animals (e.g., rabbit, chicken, juvenile sheep/goat) or part carcasses, it is still rare that whole carcasses of larger ungulates (adult sheep/goat or larger) are provided [31], which may better replicate the forces of dragging and manipulating large prey in the wild.

Whilst morphological differences between captive and wild mammals may arise due to nutritional variation [2], the nature of shape variation here (and lack of size variation) suggests that mechanical influences acting upon the jaw, cranium, and associated musculature are the primary driver. The principal jaw-closing muscles are the temporalis and masseter [66]. The masseter muscles originate at the zygomatic arch and insert on the mandible [67]. The masseter superficialis inserts on the angular process and posterior mandible [68]. A high coronoid process improves the mechanical leverage of the temporalis muscle [66] and hence increases the forces that are generated in closing the jaws for both the killing bite with the canines and chewing by the carnassials. Variation in the use of the temporalis muscle affects the height of the coronoid process during development in mammals [69,70]. The temporalis is especially important at wide jaw gape angles when the anterior teeth are used for killing and manipulating large prey in the wild [71,72]. Therefore, diets in captivity, which require lower repeated maximal biting and chewing forces to obtain, manipulate, and consume food, have likely resulted in shorter heights of the coronoid process when compared to those of wild lions and tigers. Our measurement of the postorbital bar shows a large variance and is robust to measurement error, yet does not differ between wild and captive specimens. This is curious, because of the origin of the deep masseter and superficial masseter on the ventral aspect of the postorbital bar [67], which are key muscles utilised during feeding. The skulls and mandibles of tigers and lions in zoos have been significantly affected by reduced maximal bite forces during development, which have reduced the skull musculature and mandibular height, and reduced the mechanical advantage of the jaw mechanism. Therefore, it is expected that because zoo diets result in lower forces acting upon the musculoskeletal system of the skull and mandible, big cats in zoos will have weaker bites. The extent to which this may influence fitness, if introduced into the wild under future conservation directives, is unknown, but it could be an important factor in determining the probability of survival when trying to kill and process wild prey animals.

Whilst analytical patterns of skull variation between captive and wild populations are largely in agreement across male and female lions and tigers, there are also unique patterns of variation between certain geographical groups and sexes, which are not apparent in other groups. The differentiation in sagittal crest height between captive and wild male Amur tigers is not found in tigers from Sumatra or Asian/West African lions. The inability to differentiate the captivity status of female Amur tigers by sagittal crest height or length is likely due to them possessing less pronounced sagittal crests for a given body size compared to male Amur tigers, although they are still larger than other continental tigers [16]. This is in part a scaling effect because as skull dimensions increase with body size, the surface area (a square function) of the cranium on which the masseter originates increases more slowly than jaw muscle volume (a cubic function) required for biting more powerfully [71]. It has been hypothesised that high sagittal crests develop in wild Amur tigers to support the larger temporalis muscles required to feed on frozen carcasses during winters in the Russian Far East and China [16]. These environmental conditions are not experienced by any other extant wild tiger population or by captive Amur tigers.

The lack of size differentiation between captive and wild specimens is consistent with previous studies of the lion and tiger [5], yet captive mammals, including felids, have been found to have increased skull size, body size or weights compared to their wild conspecifics, likely as a consequence of increased food availability, and decreased energy demands [1,10,19]. Because the isometric size of the cranium and mandible does not differ significantly between captive and wild lions or tigers, it is unlikely that net calorific intake (expected to be higher in captivity) or periods of calorific restriction (expected to occur in the wild) during development have affected the shape changes to the skull and mandible reported here. Most captive big cats in modern collections are fasted one or two days a week to better replicate the feeding frequencies of their wild counterparts and to promote natural feeding behaviours [28]. This may mitigate to some extent the impact of calorific excess during development and into adulthood when compared to older captive conditions with more regular feeding [1]. However, captive big cats often have large amounts of subcutaneous and deposited fat, which do not occur at such high levels in wild animals (ACK pers. obs.), so the proportion of skeletal muscle compared to fat is likely to be lower in captive big cats even if they are of similar size and mass. Nutrients in the foods may differ between captive and wild big cats and this might contribute to the variation in skull and mandible morphology. For example, horse-based diets (51% protein, 30% fat) and beef-based diets (57% protein, 28% fat) in captivity [73] may differ significantly from the nutritional composition of wild ungulate prey, and protein/fat composition of diet has been found to influence mammalian skeletal shape [40,43]. The reduction in foramen magnum height in captive lions (foramen magnum stenosis) has been attributed to vitamin deficiencies (likely vitamin A) [57] and is unlikely to be due to the mechanical properties of diet [5]. Occipital condyle breadth is also narrower in captive lions, but not tigers, which could be related to the same causes as reduced foramen magnum height and may be affected by the manipulation of large prey through the use of the surrounding neck muscles. Further investigation into variation across wild populations may answer whether foramen magnum stenosis has a potential genetic basis as suggested by [57]. 

Our results highlight measurements that are not consistently affected by captivity status across all regions of the skull, and it is likely that these measurements are less affected by bite forces, such as measurements of the orbit and skull height. These measurements may be more useful for examining evolutionary relationships between populations than those shown to vary plasticly due to diet. Mandible depth—II, measured from behind M_1_ of the mandible, was not affected by captivity status despite its position in an area of very high mechanical stress [74]. This contrasts with Mandible depth—I, measured anteriorly to the premolars, which was larger in captive lions and tigers and is also subject to high stresses during feeding. Captive big cats do not use their canines for killing or dragging large carcasses, and use their incisors less than wild big cats for feeding so that forces acting on the anterior mandible are reduced. Whilst increased stresses upon bone are expected to increase bone diameter through remodelling [75,76], increased anterior mandible depth in captive big cats due to supposed lower stresses may be a result of increased porosity and proliferation of trabecular and cortical bone where the mandible is less mechanically constrained (e.g., see [77]). A comparison of bone density in cross-sections of mandibles of wild and captive big cats could be made using CT scans to test this. The carnassial teeth (PM^4^/M_1_) are used in a similar way in both wild and captive big cats to shear through tendons, skin and muscle [78], even on smaller food items, and therefore the posterior body of the mandible experiences similar forces in captive and wild big cats.

It has been suggested that stereotypic behaviours in captivity, for example, excessive grooming, may cause variations between captive and wild samples that are independent of diet [4], but we have no data on grooming differences between wild and captive big cats. With few notable exceptions, animals tend to live longer in captivity [2,58,59]. By comparing tooth dimensions with the known ages of captive big cats, we found that tooth length decreases significantly with age (Appendix A), so it is likely that the longer lifespans of captive big cats have caused increased wear and reduced tooth sizes in comparison to their wild counterparts because dental wear and damage is progressive [79]. It is of interest that despite the relatively narrower skulls of wild lions and tigers, they have an increased cranial volume. Brain size increases in mammals in more enriched environments, and captive-bred mammals from multiple species have been found to have smaller brains [2]. While it has been suggested that the smaller cranial volume of captive lions is pathological [20], it could also be related to the development of jaw musculature impacting skull development, as well as the complexity of the environment impacting brain development. Conversely, cranial volume may be influenced by nutrition, and the Mexican wolf (*Canis lupus baileyi*) has been shown to have increased cranial volume in captivity, likely as a consequence of diet [80]. Contrary to other lions and tigers, captive Asian lions have been shown to have larger cranial volumes than wild Asian lions, which may be affected by differences in diet [81]. The plasticity of the cranial vault, foramen magnum, and the numerous skeletal structures associated with bite forces mean that great care is needed in interpreting the morphological variation of the skull in evolutionary and taxonomic contexts. This is because environmental variation in the wild across geographical space, or over time, may give rise to similar patterns of phenotypically plastic morphological variation as found between zoos and the wild, and hence obscure evolutionary patterns of morphological change. 

## 5. Conclusions

The shape of the skulls of captive and wild male and female lions and tigers differ from each other. Whilst overall skull size does not differ between captive and wild specimens, captives show an overall pattern of wider skull dimensions and shorter mandible heights. The results highlight the importance of comparing specimens from comparable taxa or populations, as morphological differences across the geographical range of each species can obscure patterns between captive and wild specimens. When similar geographical groupings are analysed, there is minimal overlap between captive and wild specimens within principal components of shape that account for between 15.4 and 34.7% of the overall variation. Thus, this paper highlights the importance of phenotypic plasticity in explaining morphological variability in the skull of the lion and tiger and suggests that the mechanical forces applied to the skull and mandible when obtaining, manipulating and consuming food best explains the patterns of shape variation. Partial- or whole-carcass feeding is common for the big cats from predominantly European institutions analysed in this study, yet the lack of overlap between captive and wild specimens suggests that the mechanics of killing, manipulating and consuming prey in the wild are not entirely replicated by carcass feeding in captivity. We speculate that the use of and load upon the anterior part of the jaws through these additional interactions with prey/food may be key to determining skull shape variation between captive and wild big cats. Differences caused by phenotypic plasticity in cranial volume, foramen magnum height, and structures associated with the mechanical forces of biting between captive and wild lions and tigers highlight the need for caution in assessing evolutionary differences between wild populations based on skull and mandible morphology. It is important to consider how these differences between captivity and the wild can be reduced for the appropriate development and welfare of captive animals, and for replicating the performance of their wild counterparts should reintroduction be considered. 

## Figures and Tables

**Figure 1 animals-13-03616-f001:**
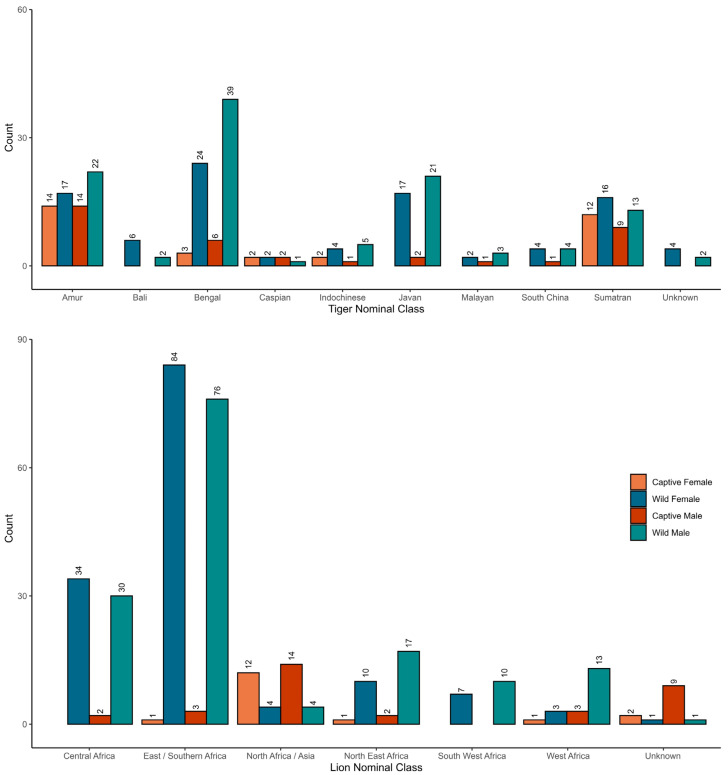
Available data for analysis grouped by sex, captivity status, and geographical origin. A total of 344 lions (184 male, 160 female) and 273 tigers (146 male, 127 female) were used in this analysis. A greater number of wild specimens was available than captive specimens of the lion (50 captive, 294 wild) and the tiger (65 captive, 208 wild).

**Figure 2 animals-13-03616-f002:**
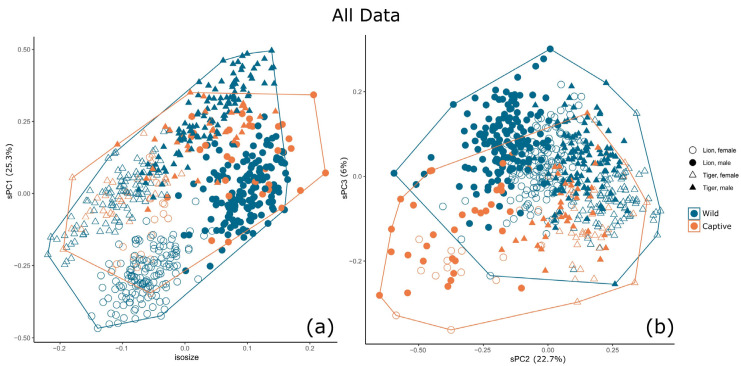
The relationship between captive and wild specimens by isosize and shape sPCs when male and female lions and tigers are analysed together; (**a**) isosize and sPC1, (**b**) sPC2 and sPC3.

**Figure 3 animals-13-03616-f003:**
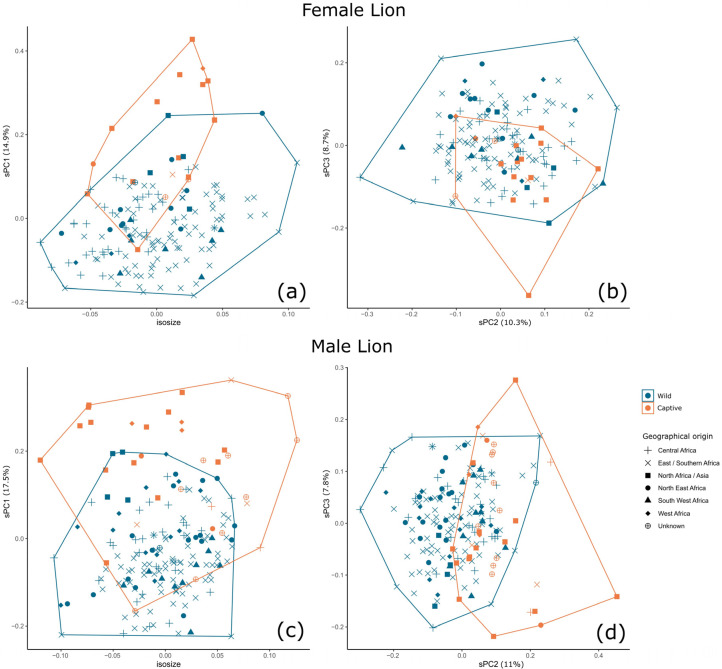
Differentiation between captive and wild specimens by isosize and sPCs for female (**a**,**b**) and male (**c**,**d**) lions. Differentiation by captivity status is apparent across sPC1 in female (**a**) and male (**c**) lions, and across sPC2 in male lions (**d**).

**Figure 4 animals-13-03616-f004:**
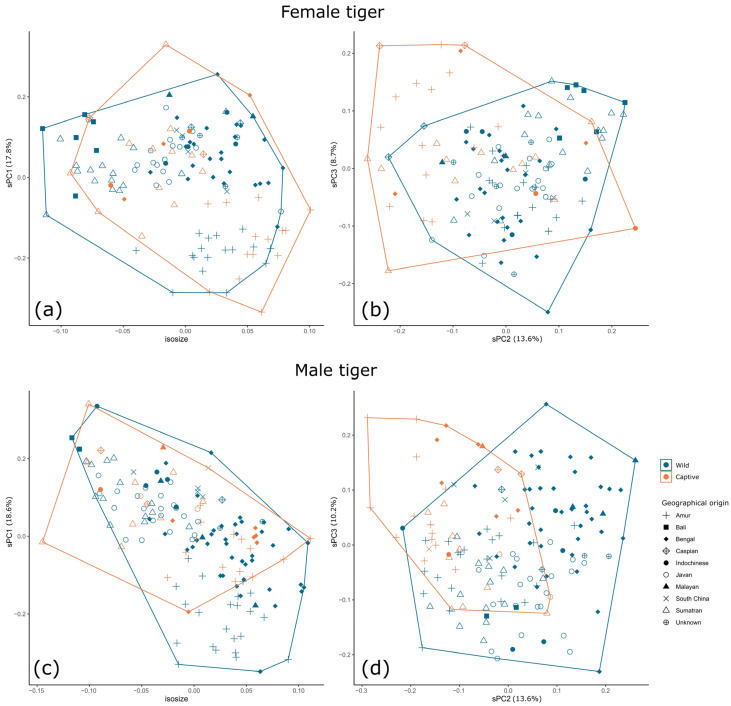
Differentiation between captive and wild specimens by isosize and sPCs for female (**a**,**b**) and male (**c**,**d**) tigers. Differentiation by captivity status is apparent across sPC2 in female (**b**) and male (**d**) tigers.

**Figure 5 animals-13-03616-f005:**
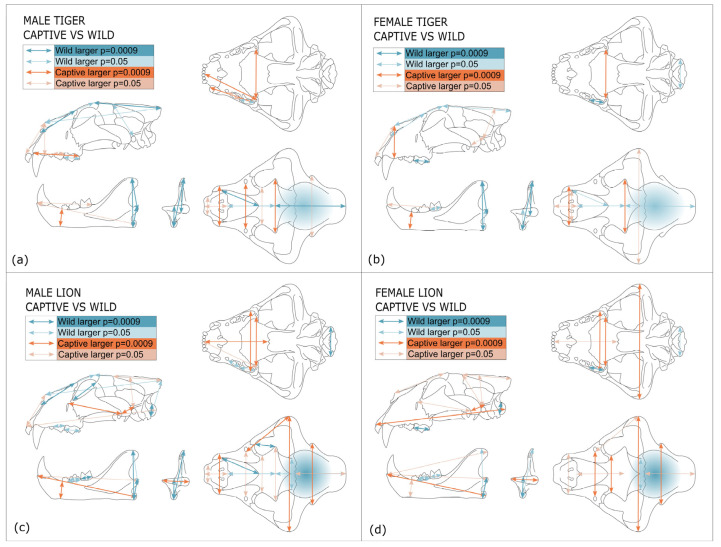
A graphical representation of measurements that are significantly different between captive and wild tigers (**a**,**b**) and lions (**c**,**d**). Cranial volume is represented by shading in the cranial region. Significance is determined by *t*-tests (Appendix A) based on values of 0.05 and after a Bonferroni correction, 0.0009.

**Figure 6 animals-13-03616-f006:**
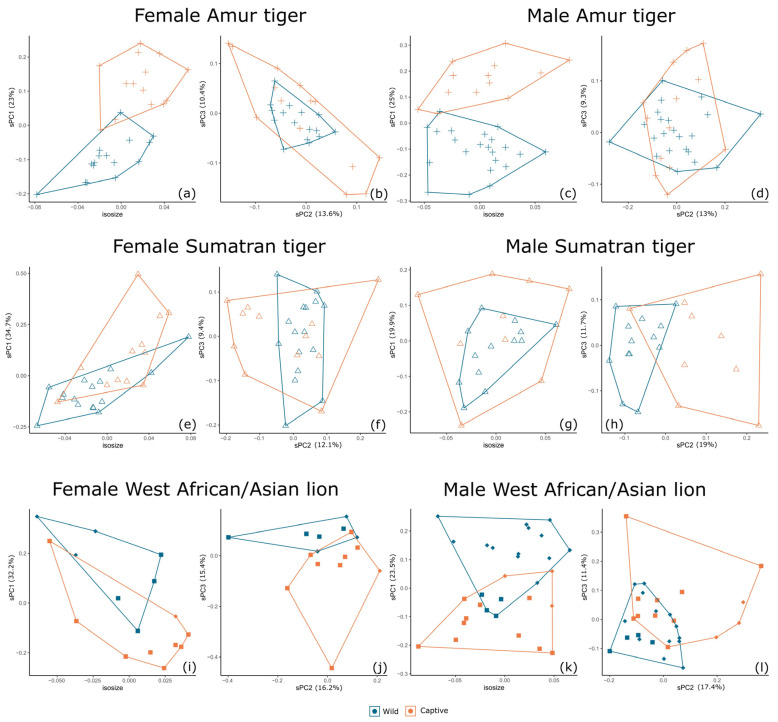
Differentiation between captive and wild specimens of Amur tiger (**a**–**d**), Sumatran tiger (**e**–**h**) and northern lion (Asian and West African lions, (**i**–**l**)) by isosize and sPCs for each sex. The shapes match the geographical groups of Figure 3 and Figure 4.

## Data Availability

Museum specimens used in this study are listed in museum_specimen_datasheet.xlsx of the Appendix A. The measurement data presented in this study are available from the corresponding authors upon reasonable request.

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
