# Peer review of "Getting to the Meat of It: The Effects of a Captive Diet upon the Skull Morphology of the Lion and Tiger"

_animals, 2023, doi:10.3390/ani13233616_

Round 1

Reviewer 1 Report

Comments and Suggestions for Authors

A generally good investigation of a very complex area.  The analysis is generally well executed and with appropriate methods. Its primary hypotheses are addressed: skull variation between captive wild skulls; indications of parameters for analysing sub-species determinations.  A number of insightful observations made in the discussion e.g. changes due to Amur tigers eating frozen carcasses; need for CT/ (dexa?) studies to assess bone density; caution on making conclusions on foramen magnum stenosis.

The main issue/ weakness revolves around the biases and variations that could be present in the set of selected skulls e.g.:

age: presumably captive animals are older than wild (hunted?) animals. 

selection via hunted / captured status: are some morphologies more likely to be hunted / kept for captivity?  Are the captive samples likely to have come from the wild? (at what age?) Are they possibly second/ third generations bred in captivity?

 year collected: it appears that range of specimen collection date varies over a century or more?  What effect would that have considering populations and environments/ captivity management is likely to have varied greatly over thus time.

I presume this detail was licking from the museum samples but the issues should be addressed in Methods and/or Discussion.

Items to consider:

147: "evolutionary history" is an inappropriate substitute for "genetics"?

165: how was repeatability measured?

169: indicate the number of missing data replaced and the criteria used for inclusion (95% confidence?)

171: Give more specific detail on how the wild/ captive status was determined.  Were specific details of the circumstances of the wild specimens determined e.g. hunted / found dead?

212: the statistical software should be referenced.

325: a brief data/ analysis of the geographical sources of the captive specimens should have been given.

Comments on the Quality of English Language

No major issues with language.

Author Response

A generally good investigation of a very complex area.  The analysis is generally well executed and with appropriate methods. Its primary hypotheses are addressed: skull variation between captive wild skulls; indications of parameters for analysing sub-species determinations.  A number of insightful observations made in the discussion e.g. changes due to Amur tigers eating frozen carcasses; need for CT/ (dexa?) studies to assess bone density; caution on making conclusions on foramen magnum stenosis.

Thank you for taking the time to review this manuscript, and for the constructive comments, which we address below.

The main issue/ weakness revolves around the biases and variations that could be present in the set of selected skulls e.g.:

age: presumably captive animals are older than wild (hunted?) animals. 

We have added a supplementary figure showing the relationship (or lack of) between size and shape principal components, and age (limited to captive individuals where age is known). As no relationship is found, it is unlikely that an age discrepancy between captive and wild big cats has influenced our findings. This is now referred to in the results and discussion (line 282, 406).

selection via hunted / captured status: are some morphologies more likely to be hunted / kept for captivity?  Are the captive samples likely to have come from the wild? (at what age?) Are they possibly second/ third generations bred in captivity?

We now address this on lines 434. Modern captive breeding programmes aim to equalise founder genetic contributions and minimise inbreeding, and historic captive specimens are most likely to have been acquired from the wild directly or be from relatively few captive generations.

 year collected: it appears that range of specimen collection date varies over a century or more?  What effect would that have considering populations and environments/ captivity management is likely to have varied greatly over thus time.

This has been addressed in the text already (line 425).

I presume this detail was lacking from the museum samples but the issues should be addressed in Methods and/or Discussion.

See above

Items to consider:

147: "evolutionary history" is an inappropriate substitute for "genetics"?

We have changed “evolutionary history” to “genetics”.

165: how was repeatability measured?

We have now addressed this: (assessed by Cooper et al., (2022) based upon the mean coefficient of variation exceeding 1% for intra- or inter- observer error)

169: indicate the number of missing data replaced and the criteria used for inclusion (95% confidence?)

We now state that missing data accounts for 3.4% of all measurements. All imputed specimens were included in the analysis, except those where captivity status or sex was unknown.

171: Give more specific detail on how the wild/ captive status was determined.  Were specific details of the circumstances of the wild specimens determined e.g. hunted / found dead?

Captivity status was determined by museum record information. The specific circumstances of wild specimens are not determined beyond location.

212: the statistical software should be referenced.

We have now referenced R at the start of this section.

325: a brief data/ analysis of the geographical sources of the captive specimens should have been given.

We have captive specimens from Europe (n=83), Africa (n=23), Asia (n=18) and Unknown (n=18). We now state this in the Approach and Methods, and have included the zoos in the museum_specimen_datasheet.

Reviewer 2 Report

Comments and Suggestions for Authors

 I found this article entitled "Getting to the meat of it: The effects of a captive diet upon the 2 skull morphology of the lion and tiger" very interesting. Although my specialty does not fully fit the topic of the article, it has been of great interest to me since it allows me to demonstrate how captivity can influence the formation of small craniological modifications in wild animals.

I think that the article could be published with minor comments. I have small suggestions that could be included in the article, which would contribute to making small improvements and giving a greater dimension to the work.

In a recent work Courtenay et al 2021, Mora et al 2023  saw from a taphonomic perspective that carnivores in captivity present differences in their dietary patterns compared to carnivores in the wild.

When carnivores eat their prey, they unconsciously produce tooth marks on the bones of their prey. Mora et al 2023 saw that carnivores in captivity bite more and leave more marks on bones than carnivores in the wild. For their part, Courenay et al 2021 observed that from a morphometric perspective, the tooth marks produced by carnivores, in the "pit" type, do not differ between wild or captive carnivores. On the contrary, in the deinte marks considered scores, it was observed that carnivores in captivity make different scores than those in the wild. The study by Courtenai et al 21 raises the possibility that stress is the cause of the scores being different between captive and wild wolves.

Although these studies refer to wolves, the authors could take into account some of the conclusions made by these investigations and in the discussion. The authors could assess or propose whether the stress of animals in captivity affects cranial differences, and If it cannot be applied in this study, it could be raised in the discussion as a line to work on in the future.

The authors may also find the work of Gidna et al 2013 useful, which also observed more bone alterations in carcasses eaten in captivity vs. freedom lions.

Although these three studies refer to the frequencies, distribution and morphology of tooth marks. The conclusions of these works complement the results of this research and support the results of this research.

references: 

Courtenay, Lloyd A., Darío Herranz-Rodrigo, José Yravedra, José Mª Vázquez-Rodríguez, Rosa Huguet, Isabel Barja, Miguel Ángel Maté-González, Maximiliano Fernández Fernández, Ángel-Luis Muñoz-Nieto y Diego González-Aguilera. 2021. "Perspectivas 3D sobre los efectos del cautiverio en la masticación de los lobos y sus marcas de dientes; implicaciones en los estudios ecológicos del pasado y del presente" Animales 11, no. 8: 2323. https://doi.org/10.3390/ani11082323

Mora, Rocío, Julia Aramendi, Lloyd A. Courtenay, Diego González-Aguilera, José Yravedra, Miguel Ángel Maté-González, Diego Prieto-Herráez, José Mª Vázquez-Rodríguez e Isabel Barja. 2022. " Ikhnos : un novedoso software para registrar y analizar modificaciones de la superficie ósea basado en documentación tridimensional" Animales 12, no. 20: 2861. https://doi.org/10.3390/ani12202861

Gidna, A.; Yravedra, J.; Domínguez-Rodrigo, M. A cautionary note on the use of captive carnivores to model wild predator behavior: a comparison of bone modification patterns on long bones by captive andwild lions. J Archaeol Sci. 2013, 40(4), 1903-1910

Author Response

I found this article entitled "Getting to the meat of it: The effects of a captive diet upon the 2 skull morphology of the lion and tiger" very interesting. Although my specialty does not fully fit the topic of the article, it has been of great interest to me since it allows me to demonstrate how captivity can influence the formation of small craniological modifications in wild animals.

I think that the article could be published with minor comments. I have small suggestions that could be included in the article, which would contribute to making small improvements and giving a greater dimension to the work.

In a recent work Courtenay et al 2021, Mora et al 2023  saw from a taphonomic perspective that carnivores in captivity present differences in their dietary patterns compared to carnivores in the wild.

When carnivores eat their prey, they unconsciously produce tooth marks on the bones of their prey. Mora et al 2023 saw that carnivores in captivity bite more and leave more marks on bones than carnivores in the wild. For their part, Courenay et al 2021 observed that from a morphometric perspective, the tooth marks produced by carnivores, in the "pit" type, do not differ between wild or captive carnivores. On the contrary, in the deinte marks considered scores, it was observed that carnivores in captivity make different scores than those in the wild. The study by Courtenai et al 21 raises the possibility that stress is the cause of the scores being different between captive and wild wolves.

Although these studies refer to wolves, the authors could take into account some of the conclusions made by these investigations and in the discussion. The authors could assess or propose whether the stress of animals in captivity affects cranial differences, and If it cannot be applied in this study, it could be raised in the discussion as a line to work on in the future.

The authors may also find the work of Gidna et al 2013 useful, which also observed more bone alterations in carcasses eaten in captivity vs. freedom lions.

Although these three studies refer to the frequencies, distribution and morphology of tooth marks. The conclusions of these works complement the results of this research and support the results of this research.

Thank you. These are useful references which we now acknowledge in the introduction.

Reviewer 3 Report

Comments and Suggestions for Authors

This study aims at describing potential morphological differences in the skulls of captive and wild tiger and lions. To achieve that goal, the authors gathered a very large dataset, which must be commended. The standard of analyses is good, and the authors explored their data quite thoroughly. However, as someone not familiar with how big cats are dealt with in zoos, I would say some information is missing in the Introduction or Materials and Methods. Furthermore, unless authors have information that was not disclosed in the paper on the pedigrees of the captive specimens, I think their argument about plasticity is, as it stands, rather weak and that differences may as well be related to drift or simply to uncontrolled relatedness of those captive cats. More details follow.

1)      My main concern with the study is the interpretation of morphological differences as necessarily plastic. Although that is of course a very plausible option, the information given by the authors do not in my opinion allow to conclude so strictly. I am no expert on big cats, but I do not believe (and sincerely hope) that today zoo specimens are all wild animals captured as young pups and raised in captivity. I believe instead that those animals and also bred in captivity, which raises the possibility of evolutionary drift, especially since bottleneck effects may be happening in the early captive generations. The more generations of captive breeding, the more likely and large this effect may be. Similarly, relatedness between captive individuals may also be playing a role. For example, Fig. 3a shows that several captive female lions which differentiate from wild specimens are from the same geographical origin. This may “only” be a biogeographical effect, but may also be due to actual relatedness between those captive females (are some of them from the same litter? Or in any way parents?). Of course, I am afraid that the authors may not have access to precise pedigrees for all captive specimens. However, if they do, they should mention it and discuss the potential effects of this. If they don’t, conclusions about plasticity should probably be toned down, and the possibility of drift / artificial selection should be at least acknowledged.

2)      Information regarding the diets of big cats in zoos is in my opinion not detailed / clear enough in the Introduction. Differences in diets are mentioned repeatedly (lines 54-55, 97-99, 106-107, 114-117), but in the Abstract, the authors suggest those differences are limited (lines 32-34). I would suggest the authors to have a specific paragraph in the Introduction mentioning which differences there are between wild and captive diets, and between different captive diets. Also completely ignored, but perhaps even more important is the diets of pups and youngs in captivity compared to wild. In addition, this paragraph should at least introduce the fact that the animals used here are mostly (but not all? The reader does not know) from European zoos (which ones?).

3)      In the same line, the origin of the animals should be detailed even more in the Material and Methods. As it stands, and unless I missed something completely, the reader only knows about which museum collections were used. I suggest the authors disclose information about the actual origin of captive specimens in the first paragraph of the Mat & Met. If there are too many different zoos and parks, then at least mention the number of captive lions, captive tigers, males, females, and add the zoos in the Supplementary spreadsheet.

4)      In a similar vein, the sample sizes shown in the caption of Figure 1 do not match those given in line 157, even when accounting for the 131 subadults removed from the dataset. Similarly, the fact that the authors state they are using 56 measurements, but only define 33 in the Supplementary is problematic (I acknowledge that the defined points with letters, but it is really confusing and unclear when the reader as to go through 3 different sections of the Supplements just to understand what measurements were used). Supp. Fig. 1 also does not show the 56 measurements, why use points with letters, rather than show all measurements as lines? Also why not include this Figure in the main text? That would make the reading more fluid in my opinion.

5)      Some methodological aspects also require clarification. I suspect the authors did things correctly, but the way they describe it may bring doubts and confusion. First, lines 181-182, the authors state that “Measurements were log-transformed and centred (Baur and Leuenberger, 2011), so that measurements are independent of size.”. However, simply log transforming and centering (by which I assume they mean they subtracted the average from all values), does not make variables independent from size. Considering the following lines, where authors mention they computed “isosize”, I would guess that they in fact computed log-shape ratios, i.e. the log of the ratio of the each measurement divided by the geometric mean (i.e. “isosize”). However, this is just a guess, and the authors should clarify this section. Second, the Bonferroni correction adjusts the significance threshold (generally known as alpha, or alpha level), not the P-values themselves. The authors should correct their terminology and the way they report their results (e.g. line 189, caption of Table S2, Figure 5 and its caption). Either give exact P-values, e.g. P = 0.00156, or relationship of P-value to the alpha threshold, e.g. P < 0.0009 or P > 0.05 for example.

Minor comments:

1)      Line 196: “the data were rescaled”, meaning they were centered to the mean of the subgroup and / or scaled to geometric size of subgroup? Please specify + see comment 5) above.

2)      Line 196-207: Indeed, geographical signal is important, and might be confounding in many analyses. Therefore, the direct interpretation of sPCs as representative of captive/wild differences for Fig. 2, 3, 4 should be toned down. For example, differences along PC1 in female lions seem to be driven quite strongly by geographical origin (e.g. the wild group is pulled “down” on PC1 by African specimens, the difference along PC2 in male tigers is driven mostly by difference between Amur specimens to the left and Malay/Bengal specimens to the right). The statement in lines 276-277 comes much too late.

3)      Lines 324-325 and 333-334: The origin of captive specimens used in this study comes way too late and is not detailed enough. This is quite confusing, as it appears to relate to the type of diet captive animals have.

4)      Lines 343-348: I find the statement that there is a differentiation and no overlap between captive and wild animals too bold here. It might be the case when looking at specific geographical regions (Fig. 6) but not really at the species level (Fig. 3-4). Furthermore, as previously mentioned, as it stands, evolutionary drift cannot be entirely ruled out, nor can the confounding effect of geographical origin. I would rephrase the following interpretation by saying that some differences are detected, but apparently not as large as when animals are fed pellets, as in N-American zoos. Small differences may in fact be quite reassuring when thinking about potential reintroductions.

5)      Lines 379-383: I assume that the “measurement of the postorbital bar” you refer to is measurements N°2 which measures the height of the anterior part of the zygomatic arch. As far as I know in Mammals, the masseter originates not from the medial, but from the ventral aspect of the arch (e.g. Druzinsky et al. 2011) and inserts on the lateral aspect of the mandible, which would create forces that are ventro-medially obliquely oriented, which can be expected to impact the height and width of the arch.

6)      Lines 387-390: Considering the overlap shown by your study in skull shape, I would expect bite force differences to be rather limited. However, it is indeed possible that muscle differences would be more pronounced.

7)      Lines 397-402: This is kind of a strange idea, if the larger sagittal crest relates to frozen preys, then wild Amur females should have it too, unless they do not feed in winter?

8)      Lines 418-423: Isn’t the reduce activity a more plausible driver of body fat differences than nutrient composition? I mean both can play a role of course, but I am surprised not to see inactivity mentioned here.

9)      Lines 490-496: The conclusions here appear to me quite bold. Since the captive breeding is not accounted for, it seems difficult to strictly relate observed differences to plasticity (but of course plasticity is a plausible candidate). As the authors mention, these captive animals are fed carcasses, so one could reverse the interpretation and say that there is a fairly large overlap between wild and captive populations (in most PCs, and at most levels of this study), which can be explained by the similarity between diets!

Author Response

This study aims at describing potential morphological differences in the skulls of captive and wild tiger and lions. To achieve that goal, the authors gathered a very large dataset, which must be commended. The standard of analyses is good, and the authors explored their data quite thoroughly. However, as someone not familiar with how big cats are dealt with in zoos, I would say some information is missing in the Introduction or Materials and Methods. Furthermore, unless authors have information that was not disclosed in the paper on the pedigrees of the captive specimens, I think their argument about plasticity is, as it stands, rather weak and that differences may as well be related to drift or simply to uncontrolled relatedness of those captive cats. More details follow.

Thank you for reviewing this manuscript. We have made changes in line with your comments as listed below:

  • My main concern with the study is the interpretation of morphological differences as necessarily plastic. Although that is of course a very plausible option, the information given by the authors do notin my opinion allow to conclude so strictly. I am no expert on big cats, but I do not believe (and sincerely hope) that today zoo specimens are all wild animals captured as young pups and raised in captivity. I believe instead that those animals and also bred in captivity, which raises the possibility of evolutionary drift, especially since bottleneck effects may be happening in the early captive generations. The more generations of captive breeding, the more likely and large this effect may be. Similarly, relatedness between captive individuals may also be playing a role. For example, Fig. 3a shows that several captive female lions which differentiate from wild specimens are from the same geographical origin. This may “only” be a biogeographical effect, but may also be due to actual relatedness between those captive females (are some of them from the same litter? Or in any way parents?). Of course, I am afraid that the authors may not have access to precise pedigrees for all captive specimens. However, if they do, they should mention it and discuss the potential effects of this. If they don’t, conclusions about plasticity should probably be toned down, and the possibility of drift / artificial selection should be at least acknowledged.

Our conclusions are based upon patterns of morphological variation that are corroborated by datasets which are genetically independent of one another (lions, tigers, Amur tigers, Sumatran tigers, West African/Asian lions) and so it is very unlikely that our main findings in shape variation are driven by genetics. We have made this more explicit within the discussion (line 434) and have argued that it is unlikely that uncontrolled relatedness has affected our results. We have added a column to the supplementary information museum specimen datasheet to show the origin of captive specimens where it is known.

Captive breeding programmes aim to minimise inbreeding and hence drift.  Where data are available, zoo Amur tigers show less inbreeding and higher genetic diversity than wild populations of Amur tiger (Henry et al. 2009). Henry, P., Miquelle, D., Sugimoto, T., McCullough, D.R., Caccone, A. and Russello, M.A., 2009. In situ population structure and ex situ representation of the endangered Amur tiger. Molecular ecology18(15), pp.3173-3184.

2)      Information regarding the diets of big cats in zoos is in my opinion not detailed / clear enough in the Introduction. Differences in diets are mentioned repeatedly (lines 54-55, 97-99, 106-107, 114-117), but in the Abstract, the authors suggest those differences are limited (lines 32-34). I would suggest the authors to have a specific paragraph in the Introduction mentioning which differences there are between wild and captive diets, and between different captive diets. Also completely ignored, but perhaps even more important is the diets of pups and youngs in captivity compared to wild. In addition, this paragraph should at least introduce the fact that the animals used here are mostly (but not all? The reader does not know) from European zoos (which ones?).

We have included more detail in the introduction to reflect the feeding practices of European zoos.

3)      In the same line, the origin of the animals should be detailed even more in the Material and Methods. As it stands, and unless I missed something completely, the reader only knows about which museum collections were used. I suggest the authors disclose information about the actual origin of captive specimens in the first paragraph of the Mat & Met. If there are too many different zoos and parks, then at least mention the number of captive lions, captive tigers, males, females, and add the zoos in the Supplementary spreadsheet.

We agree. Where possible we have added relevant information on the provenance of captive specimens to the supplementary information museum specimen data sheet.

4)      In a similar vein, the sample sizes shown in the caption of Figure 1 do not match those given in line 157, even when accounting for the 131 subadults removed from the dataset. Similarly, the fact that the authors state they are using 56 measurements, but only define 33 in the Supplementary is problematic (I acknowledge that the defined points with letters, but it is really confusing and unclear when the reader as to go through 3 different sections of the Supplements just to understand what measurements were used). Supp. Fig. 1 also does not show the 56 measurements, why use points with letters, rather than show all measurements as lines? Also why not include this Figure in the main text? That would make the reading more fluid in my opinion.

We have rechecked the numbers in the text and our spreadsheet. Our museum_specimen_datasheet.xlsx included 4 duplicate records, and our initial calculations were off by 1 specimen, which accounts for a skull of unknown sex. We have amended the datasheet and text to reflect this. We have utilized the skulls of 500 lions and 389 tigers (889 total). 131 subadults were removed, 135 specimens with >20% missing data and/or no mandibles were removed, and 6 specimens were removed where captivity status or sex was unknown.

The measurement protocol presented here has been presented in Barnett et al., (2008) and Cooper et al., (2022). Changing the diagrams to explain an existing measurement protocol would introduce unnecessary confusion in the literature. Whilst separate sections of the supplementary information are required to interpret the measurements, it is a system that allows precise repeatability through thorough reporting of the method. We have presented our results using lines (Figure 5) because this enables rapid interpretation of our results by the reader.

5)      Some methodological aspects also require clarification. I suspect the authors did things correctly, but the way they describe it may bring doubts and confusion. First, lines 181-182, the authors state that “Measurements were log-transformed and centred (Baur and Leuenberger, 2011), so that measurements are independent of size.”. However, simply log transforming and centering (by which I assume they mean they subtracted the average from all values), does not make variables independent from size. Considering the following lines, where authors mention they computed “isosize”, I would guess that they in fact computed log-shape ratios, i.e. the log of the ratio of the each measurement divided by the geometric mean (i.e. “isosize”). However, this is just a guess, and the authors should clarify this section. Second, the Bonferroni correction adjusts the significance threshold (generally known as alpha, or alpha level), not the P-values themselves. The authors should correct their terminology and the way they report their results (e.g. line 189, caption of Table S2, Figure 5 and its caption). Either give exact P-values, e.g. P = 0.00156, or relationship of P-value to the alpha threshold, e.g. P < 0.0009 or P > 0.05 for example.

The measurements are centered by subtracting the mean value. We have now stated this in the text.

We missed out a step in the analysis here. We have amended the text to reflect this, and added an additional reference to the methodology (Baur et al., 2014).

Isometric size is the geometric mean of all variables.

Minor comments:

1)      Line 196: “the data were rescaled”, meaning they were centered to the mean of the subgroup and / or scaled to geometric size of subgroup? Please specify + see comment 5) above.

The wording has been adjusted to reflect that we rescaled and centred by each data subset.

2)      Line 196-207: Indeed, geographical signal is important, and might be confounding in many analyses. Therefore, the direct interpretation of sPCs as representative of captive/wild differences for Fig. 2, 3, 4 should be toned down. For example, differences along PC1 in female lions seem to be driven quite strongly by geographical origin (e.g. the wild group is pulled “down” on PC1 by African specimens, the difference along PC2 in male tigers is driven mostly by difference between Amur specimens to the left and Malay/Bengal specimens to the right). The statement in lines 276-277 comes much too late.

We have acknowledged the potential influence of geography upon our results. When specimens of the same geographical population were analysed, the differentiation between captive and wild specimens is markedly more apparent, often accounting for the first principal component with very little overlap in shape space. Our conclusions have been based on the results as a whole, using all scales of analysis.

3)      Lines 324-325 and 333-334: The origin of captive specimens used in this study comes way too late and is not detailed enough. This is quite confusing, as it appears to relate to the type of diet captive animals have.

The non-random wild origin of specimens is explicitly presented in Figure 1 and discussed in detail within the methods section.

4)      Lines 343-348: I find the statement that there is a differentiation and no overlap between captive and wild animals too bold here. It might be the case when looking at specific geographical regions (Fig. 6) but not really at the species level (Fig. 3-4). Furthermore, as previously mentioned, as it stands, evolutionary drift cannot be entirely ruled out, nor can the confounding effect of geographical origin. I would rephrase the following interpretation by saying that some differences are detected, but apparently not as large as when animals are fed pellets, as in N-American zoos. Small differences may in fact be quite reassuring when thinking about potential reintroductions.

In this section we state that there is very low overlap or some overlap between captive and wild lions and tigers of each sub-population. This paragraph is a simple description of our results, within the results section, as they are presented in Figure 6.

5)      Lines 379-383: I assume that the “measurement of the postorbital bar” you refer to is measurements N°2 which measures the height of the anterior part of the zygomatic arch. As far as I know in Mammals, the masseter originates not from the medial, but from the ventral aspect of the arch (e.g. Druzinsky et al. 2011) and inserts on the lateral aspect of the mandible, which would create forces that are ventro-medially obliquely oriented, which can be expected to impact the height and width of the arch.

This is an error on our part. It is shown well in Hartstone Rose et al., 2012 – Bite force estimation and the fiber architecture of felid masticatory muscles. Given that the superficial and deep masseter originate from the ventral aspect of the zygomatic arch, we have deleted our speculation that mastication would predominantly affect width and not height.

6)      Lines 387-390: Considering the overlap shown by your study in skull shape, I would expect bite force differences to be rather limited. However, it is indeed possible that muscle differences would be more pronounced.

7)      Lines 397-402: This is kind of a strange idea, if the larger sagittal crest relates to frozen preys, then wild Amur females should have it too, unless they do not feed in winter?

Female Amur tigers have larger sagittal crests than other female continental tigers as found by Cooper et al., 2022. We have adjusted the text to reflect this.

8)      Lines 418-423: Isn’t the reduce activity a more plausible driver of body fat differences than nutrient composition? I mean both can play a role of course, but I am surprised not to see inactivity mentioned here.

We mention inactivity (decreased energy demands) on line 515, and why this is likely not a driving factor of the variation found here.

9)      Lines 490-496: The conclusions here appear to me quite bold. Since the captive breeding is not accounted for, it seems difficult to strictly relate observed differences to plasticity (but of course plasticity is a plausible candidate). As the authors mention, these captive animals are fed carcasses, so one could reverse the interpretation and say that there is a fairly large overlap between wild and captive populations (in most PCs, and at most levels of this study), which can be explained by the similarity between diets!

We now discuss the provenance of captive specimens in more detail in the text and in the supplementary information as per the comments above and from other reviewers. We also highlight the similarity of shape variation shown by genetically independent datasets (lions, tigers, geographical groups). We highlight that captive breeding attempts to minimize inbreeding and maximise genetic diversity and point out that many of our captive specimens would have been taken from the wild or have been from one or few captive generations. These details point to plastic rather than genetic drivers of differentiation between captive and wild populations.

Reviewer 4 Report

Comments and Suggestions for Authors

The manuscript, titled 'Getting to the meat of it: The effects of a captive diet upon the skull morphology of the lion and tiger,' addresses a fascinating topic: the disparities in skull morphology between captive and wild large felids. The primary objective of the manuscript is to explore the reasons behind the variations in the physical characteristics of the skulls and mandibles of these animals. The study is grounded in an extensive dataset of 621 specimens obtained from museum collections.

Their findings indicate that there is no significant difference in skull size between captive and wild lions and tigers. Nevertheless, disparities in skull and mandible shapes are evident, primarily in regions related to biting. These differences are mainly found in regions associated with biting, suggesting that the animals' diets have influenced the forces acting on their skulls and mandibles. Notably, captive big cats in the study predominantly consumed whole or partial carcasses, which closely mimic the mechanical properties of the diets of their wild counterparts. The text postulates that the additional demands of hunting, manipulating, and consuming large prey in the wild have contributed to the observed differences in the morphology of captive and wild big cats.

I have only a few minor comments. It would be beneficial if the authors could clarify how they centered the measurements (line 181).

While I concur with the authors' assertion that hunting and manipulating large prey in wild cats may account for the observed differences in skull morphology between wild and captive felids, I suggest that the authors consider additional factors that could contribute to these distinctions. Morphological variations might also arise from a passive selection for animals behaviorally more adapted to captivity, decreased activity levels in captive animals, the relaxation of natural selection in captive cats, or other factors like inbreeding, founder effects, and the disruption of wild reproductive patterns (Gleeson and Wilson, 2023; Hartstone-Rose et al., 2014; Kistner et al., 2021; O'Regan and Kitchener, 2005; Wisely et al., 2002).

In conclusion, this is an outstanding paper based on an impressive dataset.

Suggested references

Gleeson BT, Wilson LAB, 2023 Shared reproductive disruption, not neural crest or tameness, explains the domestication syndrome. Proc. R. Soc. B 290:20222464.

Kistner TM, Zink KD, Worthington S, Lieberman DE, 2021. Geometric morphometric investigation of craniofacial morphological change in domesticated silver foxes. Sci Rep 11, 2582.

Wisely, SM, Ososky, JJ, Buskirk, SW, 2002. Morphological changes to black-footed ferrets (Mustela nigripes) resulting from captivity. Canadian Journal of Zoology 80, 1562-1568.

2.12.0.0

Author Response

The manuscript, titled 'Getting to the meat of it: The effects of a captive diet upon the skull morphology of the lion and tiger,' addresses a fascinating topic: the disparities in skull morphology between captive and wild large felids. The primary objective of the manuscript is to explore the reasons behind the variations in the physical characteristics of the skulls and mandibles of these animals. The study is grounded in an extensive dataset of 621 specimens obtained from museum collections.

Their findings indicate that there is no significant difference in skull size between captive and wild lions and tigers. Nevertheless, disparities in skull and mandible shapes are evident, primarily in regions related to biting. These differences are mainly found in regions associated with biting, suggesting that the animals' diets have influenced the forces acting on their skulls and mandibles. Notably, captive big cats in the study predominantly consumed whole or partial carcasses, which closely mimic the mechanical properties of the diets of their wild counterparts. The text postulates that the additional demands of hunting, manipulating, and consuming large prey in the wild have contributed to the observed differences in the morphology of captive and wild big cats.

I have only a few minor comments. It would be beneficial if the authors could clarify how they centered the measurements (line 181).

The measurements are centered by subtracting the mean value. We have now stated this in the text.

While I concur with the authors' assertion that hunting and manipulating large prey in wild cats may account for the observed differences in skull morphology between wild and captive felids, I suggest that the authors consider additional factors that could contribute to these distinctions. Morphological variations might also arise from a passive selection for animals behaviorally more adapted to captivity, decreased activity levels in captive animals, the relaxation of natural selection in captive cats, or other factors like inbreeding, founder effects, and the disruption of wild reproductive patterns (Gleeson and Wilson, 2023; Hartstone-Rose et al., 2014; Kistner et al., 2021; O'Regan and Kitchener, 2005; Wisely et al., 2002).

We have highlighted the potential for the nutritional properties of diet to have affected morphology (line 527).

We discuss energetic demands of captive mammals on line 515, which is a consequence of activity levels.

We have addressed the possibility of founder effects and inbreeding in influencing our conclusions, and explained why this is unlikely (line 434).

In conclusion, this is an outstanding paper based on an impressive dataset.

Thank you for for reviewing this paper and for your constructive comments!

Round 2

Reviewer 3 Report

Comments and Suggestions for Authors

The authors have acknowledged my comments.

I still feel they could give a bit more details about the diets, notably (i) how is the diet of pups/youngs in captivity (do they breast feed on their mothers, or are they fed by caretakers? At which approximate age do they start to feed on carcasses?) and (ii) since the authors mentioned the diets lack large ungulates (I would have assumed cattle to be the main source of food), with which animal carcasses are they fed?

Also, since they seem to have good arguments showing that drift or inbreeding is limited in zoos, they should make it even more clear in text (at the moment they only have a couple short mentions).

Author Response

I still feel they could give a bit more details about the diets, notably (i) how is the diet of pups/youngs in captivity (do they breast feed on their mothers, or are they fed by caretakers? At which approximate age do they start to feed on carcasses?) and (ii) since the authors mentioned the diets lack large ungulates (I would have assumed cattle to be the main source of food), with which animal carcasses are they fed?

We now state on line 145 that weaning occurs between four and nine months in lions and tigers. We cannot say whether captive cats were bottle-fed or naturally breast-fed milk before this time, but we have acknowledged that any hand-reared cubs are less likely to have been fed whole carcasses post-weaning (line 428).

We now state on line 459: “Whilst European zoos may provide whole carcasses of small animals (e.g. rabbit, chicken, juvenile sheep/goat) or part carcasses, it is still rare that whole carcasses of larger ungulates (adult sheep/goat or larger) are provided (Kleinlugtenbelt, Burkevica and Clauss, 2023).”

Also, since they seem to have good arguments showing that drift or inbreeding is limited in zoos, they should make it even more clear in text (at the moment they only have a couple short mentions).

We have stated the following on line 432:

Because patterns of shape variation are similar between Sumatran tigers, Amur tigers, and northern lions (Figures S7-S12 of the Supplementary Information), it is unlikely that founder effects in the captive populations of each of these disparate groups have influenced our conclusions. Further, modern captive breeding programmes aim to equalise founder genetic contributions and minimise inbreeding, and historic captive specimens are most likely to have been acquired from the wild directly or be from one or relatively few captive generations.

We have included additional references to support these statements, and we now include the following sentence to reinforce the relevance of these statements:

“We therefore postulate that it is phenotypic plasticity in response to environmental differentiation (through diet), rather than genetic differentiation, that is the primary driver of the shape changes between captive and wild lions and tigers described here.”